# Histopathological profile of cervical biopsies in northern Malawi: a retrospective cross-sectional study

Paul Uchizi Kaseka ,[1] Alfred Kayira,[2] Chikondi Sharon Chimbatata,[1] Master R O Chisale ,[3] Pocha Kamudumuli,[4] Tsung-Shu Joseph Wu,[5] Balwani Chingatichifwe Mbakaya ,[6] Frank Watson Sinyiza[7]

[1]Paediatrics Department, Mzuzu Central Hospital, Mzuzu, Malawi
[2]Laboratory Department, Mzuzu Central Hospital, Mzuzu, Malawi
[3]Biological Sciences, Mzuzu University, Mzuzu, Malawi
[4]Laboratory, University of Maryland and Global Initiative Company, Lilongwe, Malawi
[5]Research, Ping Tung Christian Hospital, Pingtung, Taiwan
[6]Public Health, University of Livingstonia, Mzuzu, Malawi
[7]Clinical Department, Mzuzu Central Hospital, Mzuzu, Malawi

**Correspondence to**
Mr Paul Uchizi Kaseka;
kasekapaul2016@gmail.com

## ABSTRACT

**Objectives** According to the WHO (2014), cervical cancer is the second most common cancer in women globally. More than 85% of the global cervical cancer morbidity and mortality occur in low-income and middle-income countries and the highest risk region is in Eastern and Southern Africa. Malawi has the highest age-standardised rate of cervical cancer in the world. This study was carried out to determine the histopathological profile of cervical biopsies in a public tertiary hospital in Mzuzu, northern region of Malawi.

**Setting** A public tertiary hospital in Mzuzu, northern region of Malawi.

**Participants** This was a retrospective study of all cervical biopsy specimen reports received in a public tertiary hospital in northern Malawi over a period of 5 years from July 2013 to June 2018. Demographic, clinical and diagnostic data were obtained from original histopathology reports.

**Results** A total of 500 cervical biopsy reports were reviewed during the study period. The mean age of the patients was 41.99±12.5. Age ranged from 15 to 80 years. Cervicitis accounted for 46.0% (n=162) of the total non-malignant lesions seen, followed by cervical intraepithelial neoplasm, at 24.4% (n=86) and endocervical polyp, at 20.5% (n=72). Squamous cell carcinoma (SCC) accounted for 15.6% (n=78) of the total cervical biopsies studied and 85.7% of all total malignant lesions. Adenocarcinoma and undifferentiated carcinoma were 8.8% and 4.4%, respectively of the total malignant diagnosis. All patients with malignant lesions had HIV.

**Conclusion** Our study shows that cervicitis and SCC were most common among non-malignant and malignant cervical biopsies, respectively. Since the frequency of cervical cancer is high, there is a need to have well detailed national policies to be put in place to increase detection of preinvasive lesions in order to reduce the prevalence of cervical cancer.

## BACKGROUND

Cervical cancer, after breast cancer, is the second most common cancer in women aged 15–44 years and it is the third leading cause of cancer in females worldwide.[1] According to the International Agency for Research on Cancer estimates, there are 570 000 new

## Strengths and limitations of this study

► The study was able to determine the prevalence and associations of multiple exposures and outcomes.
► This being a retrospective cross-section study, the participants were neither deliberately exposed nor treated; thus, there were no ethical difficulties.
► This study depended on data that were entered into clinical database and not collected in a predesigned proforma as per specific requirements of the study as a result some records were excluded due to missing certain crucial information.
► Since in retrospective studies, researchers have no control over the exposure of cases versus controls, these unrecognised confounders may have influenced the results.
► The study is a single-hospital-based review and as such inadequate to draw conclusions, but it does shed some light on pathological pattern of cervical cancer in Malawi.

cases of cervical cancer annually, resulting into more than 311 000 deaths in 2018 globally.[2] Most of the global burden lies in less developed countries, with sub-Saharan Africa having the largest age-standardised incidence and mortality rates. Malawi has the highest cervical cancer incidence and mortality in the world with age-standardised rate of 75.9 and 49.8 per 100 000 population, respectively.[3] WHO estimates suggest that every year there are at least 3684 new cases of cervical cancer in Malawi and over 2314 die from the disease.[3] The exact number of cervical cancer morbidity and mortality among Malawian women, is not clear. This could partly be due to unrecorded or under-reported cases because of the pathological based cancer registry not being maintained, and also lack of a national system of death certification.[4] However, the 2010 National Population-based Cancer Registry indicates that among females, cancer of the cervix was the most common, accounting for 45.4% of

all cases followed by Kaposi's sarcoma (21.1%), cancer of the oesophagus (8.2%), breast cancer (4.6%) and non-Hodgkin's lymphoma (4.1%).[5]

The Ministry of Health and Population of the government of Malawi, through the Sexual and Reproductive Health Directorate has implemented a cervical cancer screen-and-treat programme using visual inspection with acetic acid (VIA) approach since 2004, with women between 30 and 50 years as the main target.[6] Women, no younger than 30 years, are offered three[3] free smears, with a 10 years interval in between each smear. Those screened for first time at the age of 55 or more have only one smear if the first smear is normal.[6] Cervical screening using VIA is increasingly available in local clinics through the National Cervical Cancer Control Programme.[6 7] From clinics, patients are referred to District Hospitals and/or Central Hospitals. At Mzuzu Central Hospital, women with suspected cervical cancer are managed at gynaecology and/or oncology departments. Women with low-grade squamous intraepithelial lesions require re-screening within a 12 months period, while those with high-grade lesions are referred for colposcopy.

The standard method for diagnosis of cervical precancerous lesions is histopathological examination of tissue obtained through biopsy guided by colposcopy. When abnormalities are identified, cervical biopsy confirms the diagnosis of cancer.[8] This study was carried out to determine the histopathological profile of cervical biopsies in a public tertiary hospital in Mzuzu, northern region of Malawi. The specific objectives were to determine the prevalence of both precancerous and cancerous cervical lesions; characterisation of precancerous and cancerous lesions and risk factors of cervical cancer. Understanding the histological pattern of cervical cancer could guide development of focused preventive, care and treatment guidelines to address shortfalls in the care that is provided to cervical cancer patients in Malawi. Ultimately, this could contribute to the global target of 25% reduction of premature mortality from non-communicable diseases (NCDs) by the year 2025.[9]

## MATERIALS AND METHODS
### Design, setting and population
This was a retrospective study of all cervical biopsies reports received from Kamuzu Central Hospital/University of North Carolina (KCH/UNC) pathology laboratory in Lilongwe for Mzuzu Central Hospital (MCH), the only public tertiary hospital in the northern region of Malawi. This hospital does not have a functional pathology laboratory and relies on the KCH/UNC pathology laboratory for its services. The hospital is located in the northern part of Malawi catering for a population of about 2 289 780 million people[10] and serving five government District Hospitals, three Christian Health Association of Malawi hospitals and several private hospitals and clinics.

While some women presented to the gynaecological clinic asymptomatically through screening after VIA

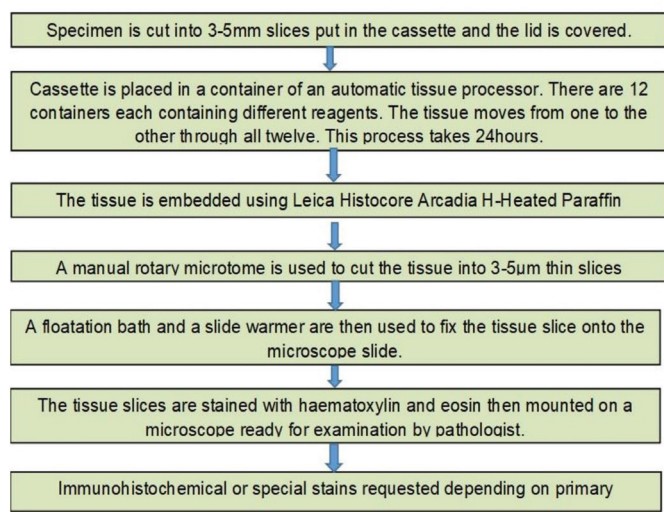

**Figure 1** Flow diagram showing the process involved in coming up with a histopathological diagnosis at Kamuzu Central Hospital/University of North Carolina laboratory in Lilongwe.

indicating a cancer, others presented with symptoms as referrals from a local clinic or other district hospitals from northern region. The most common symptoms were vaginal bleeding (often postcoital), vaginal discharge with dysuria, abdominal pain, vomiting and weight loss.

Tissue specimens were collected and preserved in 10% buffered formalin solution and then transported to KCH/UNC pathology laboratory in Lilongwe. The KCH/UNC laboratory adheres to international quality assurance standards. The flow diagram (see figure 1) shows the process involved in coming up with a histopathological diagnosis at KCH/UNC laboratory in Lilongwe.

Within the 5-year period of the study (July 2013–June 2018), a total of 2294 biopsy reports were received from KCH/UNC Histopathology laboratory. Data extracted included age, year, anatomic site, nature of specimen, clinical diagnosis, histopathological diagnosis, HIV status and whether the specimen was non-malignant or malignant. Out of 2294 biopsy reports 500 were cervical biopsy reports representing 21.8% of the total biopsy reports received. In this study a total of 500 cervical cancer pathology reports were analysed. Eleven reports which had missing demographic and clinical data or had inconclusive results were excluded. HIV status was defined as whether the patient was HIV sero reactive (positive), negative or not tested (unknown) when the biopsy was being taken. HPV status was defined as the patient sample being positive or negative on histopathology examination. HPV was diagnosed histologically through observation of dysplastic changes in the superficial cervical epithelium that are consistent with HPV infection. These changes include koilocytosis and chronic inflammation. Histologically, koilocytosis is characterised by perinuclear cavitation, enlarged nucleus with coarse chromatin making it stain dark (hyperchromasia) with Lugol's iodine solution, irregular nuclear membranes and a rim of condensed

cytoplasm around the perinuclear cavitation which gives the cells a 'halo' or cleared-out appearance around the dysplastic nucleus. Chronic inflammation on the other hand is characterised by infiltration of inflammatory cells (lymphocytes) into the cervical tissue.

### Patient and public involvement
No patient involved

### Data analysis
Data were entered in Microsoft excel 2016, validated and cleaned before importing into Stata, V.13.0 (Stata Corp) for analysis. Descriptive analyses were performed to summarise patients' sociodemographic and clinical characteristics. A multivariate logistic regression was used to estimate the magnitude of the association between predictor variables (age and HIV status) and cancer at 95% CI level.

### RESULTS
Within the 5-year period of the study (July 2013–June 2018), a total of 500 biopsy reports were received from KCH/UNC Pathology laboratory. The mean age of patients included in this study was 41.99±12.5. The age range was 15–80 years. Most of the cervical biopsies were from patients in the 31–40 age group (34.8%, n=174). Twelve per cent of the cervical biopsies were from HIV positive patients and 51% of the results had unknown HIV result (table 1).

Of all the cervical biopsies studied, 91 (18.2%) were malignant. Squamous cell carcinoma (SCC) accounted for 85.7% of all malignant lesions (table 2). Cervicitis accounted for 46.0% (n=162) of the total non-malignant lesions seen. Ten per cent of cervicitis cases and 30.2% of cervical intraepithelial neoplasm (CIN) cases had HPV, respectively (table 2).

The chances of developing cervical cancer increased with advancing age, with the risk becoming more prominent after age 60 years. Women aged 61–70 years were 6.2 times (OR=6.2, 95% CI 2.1 to 18.4) more likely to develop cervical cancer than those in the 21–30 years age category. The odds of getting cancer were higher in women living with HIV. Women with HIV were more than twice (OR=2.3, 95% CI 1.1 to 4.9) as likely to have cervical cancer as those who were HIV negative (table 3).

### DISCUSSION
To the best of our knowledge, this is the first study conducted to determine the histopathological profile of cervical biopsies in a public tertiary hospital in Mzuzu, northern region of Malawi. This study found more benign conditions (70.4%, n=352) than neoplastic malignant conditions (18.2%, n=91). This is in contrast with other studies undertaken in low-income and middle-income countries which reported more neoplastic malignant conditions than benign conditions.[8 9]

Our study found that cervicitis, an inflammatory disease comprising acute cervicitis and chronic non-specific cervicitis was the most common non-malignant condition. It accounted for 46% of all the non-malignant tumours and 32.4% of all cervical biopsies received in this study. This was consistent to a study conducted in Nigeria which had 37.5% cervicitis of all cervical biopsies included in that study.[11] Most cases of cervicitis are often due to non-specific causes or infective agents.[12] Cervicitis has been highly reported in previous studies in other countries.[12 13]

In this study CIN, a premalignant lesion ranked second in non-malignant lesions accounting for 24% of all non-malignant lesions and 17.2% of all cervical biopsies with the prevalence of CIN I, CIN II, CIN III of 34.9%, 22.1% and 43%, respectively. This finding is comparable to a study conducted in Nigeria which reported CIN constituting 15% of all cervical biopsies reviewed.[13] In that study, however, there was a reduction in the prevalence of CIN from low grade CIN to high grade CIN.[13] In our study, the high grade CIN (CIN III) was the most common. This finding indicates that invasive cervical cancer progresses

| Table 1 | Demographic profile of cervical biopsies | | |
|---|---|---|---|
| | | **Biopsies analysed** | |
| **Parameter** | | **Frequency** | **Percentage (%)** |
| Age (mean) years | 20 and below(17) | 3 | 0.6 |
| | 21–30(27) | 57 | 11.4 |
| | 31–40(36) | 174 | 34.8 |
| | 41–50(45) | 132 | 26.4 |
| | 51–60(54) | 84 | 16.8 |
| | 61–70(65) | 29 | 5.8 |
| | Above 70(75) | 7 | 1.4 |
| | Missing | 14 | 2.8 |
| HIV status | Positive | 61 | 12.2 |
| | Negative | 184 | 36.8 |
| | Unknown (not documented) | 255 | 51.0 |

**Table 2** Histopathological diagnosis of cervical biopsies

| Parameter | Frequency | Within group percentage | Percentage total |
|---|---|---|---|
| **Normal cervical tissue** | **46** | | **9.2** |
| **Non-malignant lesions** | **352** | | **70.4** |
| Cervicitis | 162 | 46.0 | 32.4 |
| Cervical intraepithelial neoplasm (CIN) | 86 | 24.4 | 17.2 |
| Endocervical polyp | 72 | 20.5 | 14.4 |
| Carcinoma in situ | 4 | 1.1 | 0.8 |
| Nabothian cyst | 4 | 1.1 | 0.8 |
| Schistosomiasis | 5 | 1.4 | 1.0 |
| Condyloma (warts) | 11 | 3.1 | 2.2 |
| Others | 8 | 2.3 | 1.6 |
| **Malignant** | **91** | | **18.2** |
| Squamous cell carcinoma | 78 | 85.7 | 15.6 |
| Adenocarcinoma | 8 | 8.8 | 1.6 |
| Undifferentiated | 4 | 4.4 | 0.8 |
| Carcinosaroma | 1 | 1.1 | 0.2 |
| **Inconclusive result** | **11** | | **2.2** |
| **Total** | **500** | | **100** |

CIN I=34.9% (30/86), CIN II=22.1% (19/86), CIN III=43.0% (37/86).
Cervicitis with HPV=10% (16/162) *CIN with HPV=30.2% (26/86).

from advanced stages of precancerous lesions. This shows that there is a need to create community awareness and strengthen early cervical cancer screening for Malawi to have better outcomes.

In the current study, 10% (n=16) of all the cervicitis cases and 30.2% (n=26) of all the CIN biopsies had HPV. This finding suggests that there is high rate of HPV infections, the causative agent of CIN in the younger age group.[13] Unfortunately, the data on population-based, age specific prevalence of HPV are not available in Malawi. However, WHO estimates that the overall HPV prevalence in Malawi is about 34%.[7] HPV infection and

precancerous lesions are usually difficult to notice and develop into full blown cancer before women realise the need to seek medical care.

The rate of malignant lesions (tumours) in our study (18.2%) is comparable to 16.2% and 12.2% in studies conducted in Benin City,[14] and in Enugu, in Nigeria,[15] respectively. Studies were done in South Africa, Saudi Arabia, India and the USA, with malignant lesions at 2.42%, 4.95%, 5.5% and 5.0%, respectively, are all at variance with the current study which explains that early cervical cancer screening helps to reduce the prevalence of cervical cancer.[16–19]

**Table 3** Association between cancer, age and HIV status

| | | Cancer (%) | Non-cancer (%) | Unadjusted OR (95% CI)* | Adjusted OR (95% CI)* |
|---|---|---|---|---|---|
| Age (years) | 20 and below | 0 (0.0) | 2 (100.0) | | |
| | 21–30 (reference) | 7 (12.3) | 50 (87.2) | – | |
| | 31–40 | 25 (14.6) | 146 (85.4) | 1.2 (0.5 to 3.0) | 1.2 (0.5 to 2.9) |
| | 41–50 | 23 (17.7) | 107 (82.3) | 1.5 (0.6 to 3.8) | 1.5 (0.6 to 3.8) |
| | 51–60 | 18 (22.2) | 63 (77.8) | 2.0 (0.8 to 5.3) | 2.2 (0.8 to 5.7) |
| | 61–70 | 13 (44.8) | 16 (55.2) | 5.8 (2.0 to 17.0) | 6.2 (2.1 to 18.4) |
| | Above 70 | 1 (20.0) | 4 (80.0) | 1.8 (0.2 to 18.3) | 1.9 (0.2 to 19.9) |
| HIV status | Negative (reference) | 26 (14.4) | 155 (85.6) | – | |
| | Positive | 15 (24.6) | 46 (75.4) | 1.9 (1.0 to 4.0) | 2.3 (1.1 to 4.9) |
| | Unknown | 50 (20.2) | 197 (79.8) | | |

Fourteen patients (4 cancer cases and 10 non-cancer cases) had no data on age and were excluded from this analysis.
*Binary logistic regression.

Among the malignant tumours, SCC was the most common histological type of cervical cancer in our study. SCC accounted for 85.7% of malignant lesions cervical cancer and was also the most common diagnosis, at 15.6% of all cervical biopsies in this study. This is consistent with findings from other studies conducted worldwide.[20–22] As Faduyile *et al* observed the high rate of SCC in Malawi and Africa could reflect the low uptake of VIA and Pap smear test which are capable of identifying dysplastic conditions before transformation to malignancy.[13] This shows that there is a need for Malawi to have well-organised cervical cancer screening and Pap smear test to reduce the prevalence of SCC.[2]

In our study, adenocarcinorma and poorly differentiated carcinorma were 8.8% and 4.4%, respectively of the total malignant diagnosis. The prevalence of adenocarcinoma and poorly differentiated carcinoma observed in this study is in contrast to other studies done in Nigeria where one study found 5.8% and 2.0% of the total malignant lesions diagnosis and another found 6.0% and 1.0%, respectively.[13 23] The high prevalence found in this study confirms a previous study findings by Chadza *et al*.[24] Chadza *et al* reported that Malawian women delay to seek medical attention due to limited knowledge on symptoms and signs, limited financial resources, limited accessibility and unavailability of cancer screening facilities hence late diagnosis. These results show that early cervical cancer screening, increased awareness, better healthcare facilities, accessibility, improved histopathological confirmatory diagnosis and early treatment by surgeons, may reduce the cervical cancer burden in Malawi.

In the current study, it has been observed that the diagnosis of cervical cancer was significantly associated with the age of the patients. The chances of developing cervical cancer increased with advancing age, with the risk becoming more prominent after age 60 years. Women aged 61–70 years were 6.2 times (OR=6.2, 95% CI 2.1 to 18.4) more likely to develop cervical cancer than those in the 21–30 years age category (table 3). It is important to note that high-quality screening programmes are important to prevent cervical cancer among unvaccinated older women.[25] None of the women in our study had received any HPV vaccine in their lifetime as the first round of the mass HPV vaccine in this country was administered in January 2019 mainly in schools, targeting 9–13 years old girls who had not yet become sexually active and the second round was given in January 2020. In Malawi, the integration of HPV vaccine programmes with adequate screening programmes in older women (aged 30–49 years) has the potential to reduce the burden of cervical cancer.

Malawi's HIV prevalence is one of the highest in the world, with 10.6% of adult population (aged 15–64) living with HIV.[26] With such a high HIV prevalence in Malawi, there is an increased risk of AIDS-defining cancers including cervical cancer. In this study, the probability of getting cancer were higher in women living with HIV. Women with HIV were more than twice (OR=2.3, 95% CI 1.1 to 4.9)as likely to have cervical cancer as those who were HIV negative (table 3). This could be attributed to the reason that HIV-infected women are more likely than HIV-uninfected women to have incident and persistent HPV cervical infections.[27] A 12 monthly cervical cancer screening, increased availability of dolutegravir based antiretroviral treatment (ART) to HIV infected women now being provided in the country, routine viral load checks (at 6 months, 12 months and every 12 months since initiation) and timely switch to second line or third line regimens to ensure viral load suppression will eventually have an impact on benign lesions progressing to invasive cervical cancer.[28] However, in this study, almost half (52.74%, n=48) of all the women diagnosed with cervical cancer had an unknown HIV status. This calls for scaling up of HIV testing in cancer screening settings for early diagnosis and ART referral and further research is warranted on barriers among HIV-infected women to seeking cancer screening services despite already being in the healthcare system.

## LIMITATIONS

This study used available programme health facility data and histopathological reports on cervical cancer. The use of health facility data has its own limitations, such as incompleteness and bias in the sense that information is obtained only from people who came to the facility and underwent biopsy, leaving out those that did not seek medical care and or were not biopsied and therefore cannot be generalised to the general population.

The other limitation of this study is that it is a single-hospital-based review and as such inadequate to draw conclusions, but it does shed some light on pathological pattern of cervical cancer in Malawi.

Finally, this is a retrospective study so we could not be able to extract details, for example, in cases where tumours were diagnosed by screening or symptoms, presence or absence of the patient comorbidities. Nevertheless, the comprehensive histopathological pattern of cervical cancer demonstrated by this study provides evidence that could be used to inform policies, strategies and intervention for prevention of cancer in Malawi.

## CONCLUSION

The SCC was the most common malignant condition and cervicitis and CIN were the most common non-malignant conditions in all the women studied. Since the frequency of cervical cancer is high, there is a need for well detailed national policies to be put in place to increase detection of preinvasive lesions, which in turn will decrease the frequency of cervical cancer in the country. The presence of chronic non-specific cervicitis in women of reproductive age is infective in origin with its attending sequelae. Intensifying screening programmes among women and provision of long term ART to the HIV infected may offer an opportunity for appropriate interventions to reduce

morbidity, mortality and reduce complications among these women.

**Acknowledgements** The authors are sincerely grateful to all the data collectors and the Laboratory manager for providing the data. The Pingtung Christian Hospital through Luke International Norway (LIN) is also appreciated for financial support.

**Contributors** PUK and FWS conceived and designed the study. AK, CSC, PK, T-SJW, MROC and BCM contributed to development of the study protocol and supervised data collection and entry. AK analysed the data and PUK drafted the manuscript. PUK is responsible for the overall content as the guarantor. All authors read and approved the final manuscript.

**Funding** The study's data collection, analysis, interpretation of data and manuscript writing was funded by Pingtung Christian Hospital, Taiwan through Luke International Norway (LIN), Malawi (Grant Number: PS-IR-108001).

**Competing interests** None declared.

**Patient and public involvement** Patients and/or the public were not involved in the design, or conduct, or reporting, or dissemination plans of this research.

**Patient consent for publication** Not applicable.

**Ethics approval** This study was approved by National Health Science Research Committee (NHSRC) as part of the main study 'Pathological profile of malignancies in northern Malawi: a retrospective study at Mzuzu Central Hospital' number 19/05/2316. Both the MCH Research and Publication Committee and the MCH Laboratory department consents were obtained for the study. The need for informed consent was exempted from the institutional review board due to the nature of the study, which involved a retrospective analysis of routinely collected data.

**Provenance and peer review** Not commissioned; externally peer reviewed.

**Data availability statement** Data are available upon reasonable request. All data relevant to the study are included in the article. All data relevant to the study are included in the article and data are available upon reasonable request from the corresponding author using the following contacts, email: kasekapaul2016@gmail.com; ORCID: 0000-0002-6651-8000.

**ORCID iDs**
Paul Uchizi Kaseka http://orcid.org/0000-0002-6651-8000
Master R O Chisale http://orcid.org/0000-0002-8301-6184
Balwani Chingatichifwe Mbakaya http://orcid.org/0000-0002-4705-9064

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
