## [Reviewer comments · BMJ Open]

ARTICLE DETAILS

TITLE (PROVISIONAL)	Histopathological Profile of Cervical Biopsies in Northern Malawi: A Retrospective Study
AUTHORS	Kaseka, Paul ; Kayira, Alfred; Chibatata, Chikondi; Chisale, Master; Kamudumuli, Pocha; Wu, Tsung-Shu; Mbakaya, Balwani; Sinyiza, Frank

VERSION 1 – REVIEW

REVIEWER	Alcántara-Quintana, Luz UASLP
REVIEW RETURNED	03-Mar-2021

GENERAL COMMENTS	It is necessary to do this type of retrospective work, to publicize the cervical characteristics of the patients. As well as unifying criteria between pathologists.
--

REVIEWER	Bleeker, Maaïke VU University Medical Centre Amsterdam, Pathology
REVIEW RETURNED	02-May-2021

GENERAL COMMENTS	The paper Histopathological profile of cervical biopsies in Northern Malawi: A Retrospective Study aims determination of the histopathological outcome of 500 cervical biopsies over a period of 5 years in Northern Malawi. Information on the distribution on cervical diseases in this country is very important as the incidence of cervical cancer is one of the highest in the world. The data which are presented are moderate to clearly presented though there are multiple misspellings throughout the paper and there are several issues that deserves more attention or should be improved. Specific points: Abstract 1) Give more clearly (1-2 sentences) the rationale behind the research aim of this study. To save words in the abstract, the 11 excluded reports can be left out here (and moved to the result section). This is well presented in the 'strengths'2) Include not only the SCC but also the other malignant tumors (adenocarcinoma etc)3) 'All malignant tumours had HIV' is a strange sentence as HIV is not mentioned before and tumours do not have HIV. Patients can have HIV. If you want to give this information then you should also give an impression for the HIV status in the population studied and in the different disease groups.4) Limitation: I think that the first point mentioned is not a limitation as this is a reflection of the current situation in Malawi. Although this situation limits the success of the cervical cancer reduction, this is not primarily a limitation of the study.
--

	Introduction 5) The authors should give some more information on how current screening in Malawi is organized. Is VIA used (see-and-treat management)? Are cervical scrapes performed? Methods 6) It should be clearly stated if the 500 biopsies analyzed were of 500 different patients or not. Do these also include multiple reports (biopsy and resection or LLETZ) from one patient? 7) Could you elaborate a bit more on how the cervical samples were retrieved? Through which pathways do patients arrive in the clinic? Are these women all walk-ins with gynecological complaints? Or is there any way of screening with pap-smears after which women are referred for colposcopy with cervical biopsy? 8) It was mentioned that beside age and year, also the nature of the specimen and the clinical diagnosis was recorded? Could this be stated more clearly? Nature (it were all cervical biopsies or do they mean for instance a clockwise indication, for instance biopsy at 12 o'clock?); Clinical diagnosis (what is meant? Can they present these results), 9) How was the HIV status and the HPV status defined? Results 10) T1 can be completed with all clinical and demographical data (as summarized in the methods). Include also mean age with subheadings of the age-groups 11) T2 shows the histopathological diagnosis of cervical biopsies. The outline of the words Malignant and inconclusive seems off, should maybe be the same as normal cervical tissue and nonmalignant tissue. Adenocarcinoma (misspelled adenocarcioRma) should probably be outlined the same as squamous cell carcinoma. 12) In the statistical analysis you report using Chi square and Fisher's exact test to look for significant associations, but you do not report any comparisons / p-values? This should be added. 13) What is the total of patients included for the analysis in T3? In T 2 you report 91 malignant cases, but the sum of $7+25+23+18+13+1 = 87$ (for the age analysis). Same for the non-cancer cases, which is $46+352 = 398$ in T2. In the age analysis of T3 the sum of non-cancer cases is $n = 388$. Was age data not available for all cases or why are there cases missing? Please clarify this point. 14) T2 mentions only 30.2% HPV positivity in CIN lesions. This seems quite low. How is these HPV positivity rate in the different CIN groups? Do the high-grade CIN lesions have a higher HPV positivity rate? Or could there be another reason why the HPV positivity rate is so low? See also comment 9: I think when HPV is scored as 'viral changes' at histopathology that this information is not informative when you want to compare noncancer with cancer cases. 15) T2: check for misspellings (Nabothian cyst, Condyloma, carcinoma, carcinosaroma). 16) T2: Poorly differentiated is not a category (it goes always with either SCC or adenocarcinoma), if not clear consider either the term undifferentiated or carcinoma NOS 17) Are gynaecologists and/or pathologists involved in this paper? 18) Present consistently with one decimal (instead of alternative 1 or 2)
--	---

	Discussion 19) In the discussion section you refer to reference number 11, a study in which the HPV prevalence in the united states among women aged 14 to 59 was evaluated. However, a big difference with your data is that this prevalence is evaluated within the general population, and not within a population of women with CIN. Interpretations should be more careful, see also comment 9 and 14. 20) Page 10, line 32: nonmalignant tumour. The term tumour is not justified for CIN I, II or III 21) Page 11, line 26: 85% SCC and 15% adenocarcinoma is more or less worldwide (and not only in Malawi, Nepal and Pakistan). It is not clear why a comparison with these 2 countries is made.
--	--

REVIEWER	Lisa, Mona All India Institute of Medical Sciences - Deoghar, Department Pathology
REVIEW RETURNED	10-Aug-2021

GENERAL COMMENTS	There are some spelling and grammatical errors which I have marked in the file attached. I have also mentioned in the sticky notes some of my queries. please answer them in the manuscript and resubmit. DOI should be added to the references.
--

VERSION 1 – AUTHOR RESPONSE

Reviewer: 1 Dr. Luz Alcántara-Quintana, UASLP	It is necessary to do this type of retrospective work, to publicize the cervical characteristics of the patients. As well as unifying criteria between pathologists.	Thank you so much for your comment.
Reviewer: 2 Dr. Maaike Bleeker, VU University Medical Centre Amsterdam	The paper Histopathological profile of cervical biopsies in Northern Malawi: A Retrospective Study aims determination of the histopathological outcome of 500 cervical biopsies over a period of 5 years in Northern Malawi. Information on the distribution on cervical diseases in this country is very important as the incidence of cervical cancer is one of the highest in the world. The data which are presented are moderate to clearly presented though there are multiple misspellings throughout the paper and there are several issues that deserves more attention or should be improved. Specific points:	Thank you so much for your comment
	Abstract 1) Give more clearly (1-2 sentences) the rationale behind the research aim of this study. To save words in the abstract, the 11 excluded reports can be left out her (and	Thank you so much for your comment. See details on page 1

	moved to the result section). This is well presented in the 'strengths'	
	2) Include not only the SCC but also the other malignant tumors (adenocarcinoma etc)	Thank you so much for your comment. A sentence on adenocarcinoma and undifferentiated carcinoma has been added. See details on page 2.
	3) 'All malignant tumours had HIV' is a strange sentence as HIV is not mentioned before and tumours do not have HIV. Patients can have HIV. If you want to give this information than you should also give an impression for the HIV status in the population studied and in de different disease groups.	Thank you so much for your comment. This sentence has been corrected to reflect patients and not biopsies. See details on page 2.
	4) Limitation: I think that the first point mentioned is not a limitation as this is a reflection of the current situation in Malawi. Although this situation limits the success of the cervical cancer reduction, this is not primarily a limitation of the study.	Thank you so much for your comment The limitation has been removed. See details on page 2
	Introduction 5) The authors should give some more information on how current screening in Malawi is organized. Is VIA used (see-and-treat management)? Are cervical scrapes performed?	Thank you so much for your comment. Information on how current screening in Malawi is organized has been added. See details on page 3 and 4.
	Methods 6) It should be clearly stated if the 500 biopsies analyzed were of 500 different patients or not. Do these also include multiple reports (biopsy and resection or LLETZ) from one patient?	Thank you so much for your comment In this study a total of 500 individual patient cervical cancer pathology reports were analysed. See details on page 4 to 6.
	7) Could you elaborate a bit more on how the cervical samples were retrieved? Through which pathways do patients arrive in the clinic? Are these women all walk-ins	Thank you so much for your comment

	with gynecological complaints? Or is there any way of screening with pap-smears after which women are referred for colposcopy with cervical biopsy?	Pathway has been described. See details on page 4 to 6.
	8) It was mentioned that beside age and year, also the nature of the specimen and the clinical diagnosis was recorded? Could this be stated more clearly? Nature (it were all cervical biopsies or do they mean for instance a clockwise indication, for instance biopsy at 12 o'clock?); Clinical diagnosis (what is meant? Can they present these results),	Thank you so much for your comment. This has now been clarified in the methods. See details on page 6.
	9) How was the HIV status and the HPV status defined?	HIV status was defined as whether the patient was HIV sero reactive (positive), negative or not tested (unknown) when the biopsy was being taken. HPV status was defined as the patient sample was being positive or negative upon histopathology examination. These details have been provided in the methods. See details on page 6.
	Results 10) T1 can be completed with all clinical and demographical data (as summarized in the methods). Include also mean age with subheadings of the age-groups	Thank you so much for the comment. More information has been added to methods section to clarify what happened. Mean age for all age groups has however been added in T1. See details on page 7
	11) T2 shows the histopathological diagnosis of cervical biopsies. The outline of the words Malignant and inconclusive seems off, should maybe be the same as normal cervical tissue and nonmalignant tissue. Adenocarcinoma (misspelled adenocarcioRma) should	Thank you very much for your comment. The outline of the words have been corrected.

	probably be outlined the same as squamous cell carcinoma.	Adenocarcinoma spelling has been corrected. See details on page 8.
	12) In the statistical analysis you report using Chi square and Fisher's exact test to look for significant associations, but you do not report any comparisons / p-values? This should be added.	Thank you so much for your observation. The Ch2 of Fisher's exact test were left there in error. They were initially included in the methods section of the proposal. At that time the thinking was that we would find many variables from the records that we reviewed. In that case we planned to use a Chi2 or Fisher's exact test as appropriate to explore and isolate variables that were significantly associated with cancer at alpha (significant level) of 0.05 or less. Only those variables demonstrating a significant association with cancer at 95% confidence level would be carried forward for analysis in a multivariable logistic regression so as to quantify the association. Unfortunately, data was available only on two predictor variables – age and HIV status. Because these variables were too few we just included both of them in the logistic regression at once. We have thus removed Chi2 or Fisher' exact test from

		the test. Please see page number 6.
	13) What is the total number of patients included for the analysis in T3? In T 2 you report 91 malignant cases, but the sum of $7+25+23+18+13+1 = 87$ (for the age analysis). Same for the non-cancer cases, which is $46+352 = 398$ in T2. In the age analysis of T3 the sum of non-cancer cases is $n = 388$. Was age data not available for all cases or why are there cases missing? Please clarify this point.	Thank you for your observation. The missing cases have now been accounted for. See footnote of Table 3 on page 10.
	14) T2 mentions only 30.2% HPV positivity in CIN lesions. This seems quite low. How is these HPV positivity rate in the different CIN groups? Do the high-grade CIN lesions have a higher HPV positivity rate? Or could there be another reason why the HPV positivity rate is so low? See also comment 9: I think when HPV is scored as 'viral changes' at histopathology that this information is not informative when you want to compare noncancer with cancer cases.	HPV positivity rate in the different CIN groups (i.e. CIN I, CIN II, CIN III) has been provided as a footnote of T2 on page 8. CIN III had higher HPV positivity of 43.0%
	15) T2: check for misspellings (Nabothian cyst, Condyloma, carcinoma, carcinosaroma).	Thank you very much for the comment. Spellings for Nabothian cyst, Condyloma, carcinoma, carcinosaroma have corrected. Details are on page 8.
	16) T2: Poorly differentiated is not a category (it goes always with either SCC or adenocarcinoma), if not clear consider either the term undifferentiated or carcinoma NOS	Thank you very much for the comment. The term "poorly differentiated" has been replaced with "undifferentiated". Refer to page 8
	17) Are gynaecologists and/or pathologists involved in this paper?	Thank you very much for the comment. No gynaecologists and/or pathologists is involved in this paper
	18) Present consistently with one decimal (instead of alternative 1 or 2)	Thank you very much for the comment. One

		decimal has been used throughout the paper
	Discussion 19) In the discussion section you refer to reference number 11, a study in which the HPV prevalence in the united states among women aged 14 to 59 was evaluated. However, a big difference with your data is that this prevalence is evaluated within the general population, and not within a population of women with CIN. Interpretations should be more careful, see also comment 9 and 14.	Thank you very much for the comment. Reference has been removed. See details on pages 11 and 12.
	20) Page 10, line 32: nonmalignant tumour. The term tumour is not justified for CIN I, II or III	Term “tumour” has been replaced with “lesion”. See details on pages 11
	21) Page 11, line 26: 85% SCC and 15% adenocarcinoma is more or less worldwide (and not only in Malawi, Nepal and Pakistan). It is not clear why a comparison with these 2 countries is made.	Reference has been removed. See details on page 12.
Reviewer: 3 Dr. Mona Lisa, All India Institute of Medical Sciences - Deoghar	There are some spelling and grammatical errors which I have marked in the file attached. I have also mentioned in the sticky notes some of my queries. please answer them in the manuscript and resubmit. DOI should be added to the references.	Thank you very much for your comments. Spellings have been corrected. Issues raised in the manuscripts have been responded to accordingly. DOI has been added to references. See details on page 15-17.

VERSION 2 – REVIEW

REVIEWER	Lisa, Mona All India Institute of Medical Sciences - Deoghar, Department Pathology
REVIEW RETURNED	17-Oct-2021
GENERAL COMMENTS	In the method section please mention how the HPV status was assessed in detail - whether only histopathological features were seen (if yes, describe in a few lines) or immunohistochemistry for HPV was applied.

VERSION 2 – AUTHOR RESPONSE

HPV was diagnosed histologically through observation of dysplastic changes in the superficial cervical epithelium that are consistent with HPV infection. These changes include koilocytosis and chronic inflammation. Histologically, koilocytosis is characterized by perinuclear cavitation, enlarged nucleus with coarse chromatin making it stain dark (hyperchromasia) with Lugol's iodine solution, irregular nuclear membranes and a rim of condensed cytoplasm around the perinuclear cavitation which gives the cells a 'halo' or cleared-out appearance around the dysplastic nucleus. Chronic inflammation on the other hand is characterized by infiltration of inflammatory cells (lymphocytes) into the cervical tissue.